# A Study of Characteristics of Aluminum Bronze Coatings Applied to Steel Using Additive Technologies

**DOI:** 10.3390/ma13020461

**Published:** 2020-01-18

**Authors:** Marina Samodurova, Nataliya Shaburova, Olga Samoilova, Liudmila Radionova, Ramil’ Zakirov, Kirill Pashkeev, Vyacheslav Myasoedov, Ivan Erdakov, Evgeny Trofimov

**Affiliations:** 1Department of Metal Forming, South Ural State University, Lenin prospect 76, Chelyabinsk 454080, Russia; samodurovamn@susu.ru (M.S.); radionovalv@susu.ru (L.R.); 2Department of Materials Science, Physical and Chemical Properties of Materials, South Ural State University, Lenin prospect 76, Chelyabinsk 454080, Russia; samoylova_o@mail.ru (O.S.); tea7510@gmail.com (E.T.); 3Scientific and Educational Center “Engineering”, South Ural State University, Lenin prospect 76, Chelyabinsk 454080, Russia; zakirovra@susu.ru; 4Resource Center for Special Metallurgy, South Ural State University, Lenin prospect 76, Chelyabinsk 454080, Russia; pashkeevki@susu.ru (K.P.); vmyasoedov74@mail.ru (V.M.); 5Foundry Department, South Ural State University, Lenin prospect 76, Chelyabinsk 454080, Russia; erdakovin@susu.ru

**Keywords:** protective coating, aluminum bronze, additive technologies, microstructure, microhardness, tribological characteristics

## Abstract

The influence of laser power on the microstructural, strength, and tribological characteristics of aluminum bronze coatings applied to steel by laser cladding was studied. It was found that with an increase in laser power, the morphology of the coating surface becomes more uniform without extreme height differences. This study revealed that the coating microstructure corresponds to that of a composite material and consists of a bronze matrix and iron dendrites of different sizes (depending on the laser power). Such a microstructure affects the microhardness indices, which have a scatter of values over the coating thickness. There is a diffusion zone at the steel–bronze interface, which promotes adhesion of the matrix and coating materials. According to the results of tribological tests, the dry friction coefficient for the studied samples is in the range of 0.389–0.574.

## 1. Introduction

Copper-alloy details in friction pairs, as well as copper-based coatings applied to steel details of friction units, are widely used in automotive and mechanical engineering due to their tribological properties [1,2,3,4,5,6].

Along with electric-arc [7] and plasma spraying [8], the method of applying bronze coatings using additive technologies is considered one of the most promising nowadays. However, there are few papers studying additive application technologies and characteristics of the resulting coating [9,10,11,12]. Thus, Schmidt et al. [9] described the process of applying aluminum bronze on stainless steel tools using the direct laser deposition (DLD) method. The best results were obtained with a laser power of 3000 W, a distance of 15 mm from the nozzle to the steel surface, and a small laser beam diameter. The study of mechanical characteristics in [9] showed that the microhardness of a bronze coating obtained using additive techniques is 1.5–2 times greater than the microhardness of a steel substrate. Liu et al. [10] described the shaped metal deposition (SMD) technique of applying bronze to low-carbon steel and was good bronze–steel adhesion, which was achieved, among other things, by the diffusion of silicon into iron with the formation of solid solutions. Freiße et al. [11] studied tribological characteristics of aluminum bronze coatings applied to steel by laser cladding. It was noted that the characteristics of the coating may depend on laser power. Wang et al. [12] compared characteristics of aluminum bronze coatings, applied to steel by plasma spraying in the first case and laser cladding in the second. It was noted that the microstructure of the laser deposited coating is more uniform and consists of finer microstructural components in comparison with plasma spraying. The loss on abrasion and friction coefficient depending on load in plasma coatings have a sudden change, so the stability of such a coating is worse than that of a laser cladded one.

Thus, the data available in the literature do not allow a generalized assessment of the characteristics of a bronze coating on steel depending on the laser power.

The purpose of this paper is a comprehensive study of the microstructural, strength, and tribological characteristics of aluminum bronze coatings deposited using additive technologies on structural steel, and further establishing the dependence of the coating properties on the power of the laser used.

## 2. Materials and Methods

As the coating material, we chose aluminum bronze powder (GTV GmbH, Luckenbach, Germany) (particles with sizes from 40 to 100 micrometers) of the following composition (weight %): 8.5–10% Al, 2–4% Fe, 0.5% Mn, 0.25% Si, 0.6% Sn, and bal. Cu. According to [13], this bronze has good wear resistance indices. Low-alloyed structural steel used as a substrate had the composition (weight %): 0.34% C, 0.38% Si, 0.68% Mn, 1.47% Cr, 1.53% Ni, 0.25% Mo, 0.017% S, <0.035% P, and bal. Fe. The substrate material was used in the annealed state.

To apply the coating, a laser metal-cladding unit FL-Clad-R-4 was used (IPG IRE-Polus, IPG Photonics Corporation, Fryazino, Russia). The main unit of the complex can be represented as follows: (1) 4-kW laser head with an ytterbium fiber-optics laser with wavelength 1065–1075 nm (LS-4), the heating mode is continuous; (2) KUKA R-120 six-axis robot-equipped manipulator combined with KUKA DKP-400 double-axis positioning element; (3) TWIN-10-CR-2 powder feeder with a four-axis powder feed module; (4) process chamber–a metal cylinder with the diameter of 600 mm and the length of 1100 mm. The trajectory of laser path was linear. Surfaced strips of coating partially overlapped each other. The overlap width was 1.8 mm. The coating was carried out under shielding gas (argon), the flow rate of which was 20–22 L/min. Technological parameters of the process are given in Table 1.

The structure was studied on transverse sections of the samples, using the Axio Observer D1.m optical (Carl Zeiss Microscopy GmbH, Jena, Germany) inverted metallographic microscope equipped with the Thixomet Pro software (Thixomet Pro, Thixomet Company, Saint Petersburg, Russia) and hardware system for image analysis. The study of the transverse sections was also performed using the JEOL JSM-7001F scanning electron microscope (SEM) (JEOL, Tokyo, Japan) with Oxford Instruments energy dispersive x-ray spectrometer (EDS) (Oxford Instruments, Abingdon, UK) for quantitative and qualitative X-ray microanalysis (XRMA). The microstructure of the substrate was studied after etching in a 4% solution of nitric acid in ethanol.

The microhardness of the coating and substrate (for steel in the initial (uncoated) annealed state) was measured on the FM-800 microhardness (Future-Tech Corp, Kawasaki, Japan) tester with a load of 25 g. The number of microhardness measurements varied from 10 to 100 both across and along the obtained coating (the number of measurements depended on the thickness of the coating).

Tribological tests were carried out on the II-5018 friction machine (Tochpribor, Ivanovo, Russia) according to the scheme “movable roller–stationary roller (test material)”. The material of the machine roller was 5140H steel, the radius of the both rollers was 45 mm, the thickness was 10 mm, the rotation speed was 300 rpm, the tests were carried out under dry friction conditions in at least three sections for each of the samples. Tribological tests were carried out at room temperature. Before testing, the friction surface was ground on an abrasive material with a grain size of 9 μm. The hardness of counter material was 490 HV. Wear loss was determined by measuring the thickness loss of the coating. The test time was determined by the length of the path traveled, which for the test samples was 100 m. Wear rate was defined as the ratio of wear loss to the distance traveled.

## 3. Results

### 3.1. Morphology and Microstructure

The appearance of the resulting coating is shown in Figure 1. As the laser power increases (from Figure 1a to Figure 1e), the morphology of the coating surface changes: it covers a large area of the substrate material, becomes denser and smoother, and the roughness of the coating decreases.

When studying the transverse sections of experimental samples under the optical microscope, the thicknesses of the layer of deposited aluminum bronze were determined. For samples No. 1–3, due to the unevenness of the obtained coating, two cycles of laser surfacing were performed and the average thickness of the deposited bronze was 95 μm for sample No. 1, 1115 μm for sample No. 2 (the most uneven coating thickness with a minimum value of about 100 μm), and 1896 μm for sample No. 3. For samples No. 4 and 5 (with a uniform layer of the bronze coating in one surfacing cycle), the average thicknesses of the coatings were 936 and 1047 μm, respectively. In Figure 2, profiles of the coatings of samples No. 2 and 5 are shown to be compared.

Using SEM, we studied the contact points of steel and the bronze coating on the transverse sections of the samples, as well as the microstructure of the resulting surfacing (Figure 3, Figure 4 and Figure 5).

The transverse sections of samples after etching were studied using optical microscopy to determine the thermal effect on the microstructure of the substrate material–steel (Figure 6). The size of a heat-affected zone (HAZ) was estimated. For sample No. 1, the HAZ width was about 800 μm, for No. 2 it was 1167 μm, No. 3 had 1191 μm, No. 4 had 1473 μm, and No. 5 had 1892 μm.

### 3.2. XRMA Analysis

Average compositions of the obtained composite coating, according to X-ray microanalysis data, are given in Table 2. With an increase in the laser power, the iron concentration in the coating volume increases, and the copper concentration falls. This is consistent with the observed microstructure, since with increasing laser power, the amount of iron dendrites in the coating microstructure increases.

In the course of this study, in the transverse sections corresponding to the contact point of steel and the bronze coating, the X-ray microanalysis was performed, which confirmed the diffusion of copper and aluminum from bronze to steel (Figure 7). The study also found diffusion of aluminum and copper into iron dendrites permeating the microstructure of the coating itself.

### 3.3. Microhardness HV

The microhardness of the deposited coating for test samples is characterized by a spread in the obtained HV values over the coating width, which is in good agreement with the observed microstructure.

Since sample No. 1 is characterized by low fusion of the substrate material with the coating material, the microhardness index was only 121 HV in the upper part of the coating. When measuring the microhardness in the direction of the substrate, the HV index also increases and reaches a maximum of 270 HV in the zone of iron dendrite formation (close to the contact zone, see Figure 4a).

For samples No. 2–5, the microstructure of the coating over the entire width corresponds to the structure of a composite material. However, as noted above, as the laser power increases, the microstructure becomes more uniform, which can also be traced by microhardness. So, for sample No. 2, the range of values is from 173 to 382 HV; No. 3 is from 191 to 368 HV; No. 4 is from 216 to 347 HV; No. 5 is from 232 to 328 HV.

### 3.4. Tribological Investigations

The results of tribological tests are given in Table 3. The friction coefficient is given for different loads of 200, 300, and 400 N; wear rate is given for a sliding distance of 70 m. For sample No. 1, such features of the applied coating as non-uniformity and discontinuity did not allow testing without significant error. So, the results for this sample in Table 3 are not shown.

## 4. Discussion

It can be seen that, using the low-power 1000-W laser (Figure 2a), we obtain a bumpy coating whose thickness (on the ridge and in the cavity) can vary by almost five times. When the laser power doubles (Figure 2b), the coating becomes more uniform in thickness (difference does not exceed 16%). A similar trend in coating technology using additive technology was also noted by Schmidt et al. [9].

It was noticeable that during surfacing, fusion (mixing) of the substrate material (dark gray areas) with the coating material (light gray areas) occurred in the contact zone (see Figure 3 and Figure 4). This led to the formation of a microstructure that is typical of a composite material in which a bronze coating plays the role of a matrix, and iron dendrites are a reinforcing component. Segments enriched with iron are found in both the contact zone and in the thickness of the coating. Such a microstructure is not observed for other methods of applying a bronze coating: in the works by Zhang et al. [7] and Alam et al. [8] obtained coating made of Cu–10Al–1Fe bronze was seen to exhibit a layered nature without precipitation of additional iron particles. However, such a “mixing” in the contact zone is observed by Liu et al. [10] when applying silicon bronze to steel by the shaped metal deposition method.

Precipitation of excess iron in our tested samples, both in the contact zone and in the thickness of the coating, can be explained as follows. When the coating is applied by laser heating, not only the sprayed aluminum bronze powder is melted, but also the top layer of the steel substrate, as a result of which a melt belonging to the Cu–Fe system is formed in the contact zone. Note that the Cu–Fe system is characterized by the absence of intermetallic compounds, unlimited solubility in the liquid state, and limited solubility in the solid state [14,15]. Thus, during the subsequent cooling of the molten bath, which proceeds from the substrate toward the outer surface, dendrites of more refractory iron begin to grow first, and only then the crystallization of the main coating occurs.

With an increase in laser power, the thickness of the melted substrate increases, and the coating is more intensively enriched with iron. The microstructure of the coating under increasing laser power becomes more uniform, but, at the same time, coarser: iron dendrites penetrating into bronze become larger while copper-based interdendritic matrix occupies a smaller area.

An interesting microstructure characterized by the appearance of the so-called “spherules” of iron is presented in Figure 5a,d. Such precipitates are typical of processes involving a supercooled liquid phase as well as for non-equilibrium crystallization in the Cu–Fe system [16,17,18,19,20]. The precipitation of spherical iron particles in the tested samples is consistent with experimental data obtained by Freiße et al. [11] and Dai et al. [21] on the microstructure of an aluminum bronze coating obtained by laser cladding.

In samples No. 3–5, single cracks, starting in the contact zone, and going deep into the substrate metal no more than 300 μm, are observed. The cracks are not hollow, but filled with bronze, which should reduce the negative consequences of their formation.

According to the results of optical microscopy (see Figure 6), the heat-affected zone (HAZ) in steel has Widmanstätten microstructure that is typical for accelerated cooling of overheated steel. The observed microstructure of the HAZ is comparable with the microstructure of welded joints. The obtained data on the HAZ of experimental samples can be compared with the results by Freiße et al. [11], where in the zone bordering the deposited bronze coating, the microhardness of the steel shows a maximum and is about 600 HV, which indirectly indicates the formation of a heat-affected zone with nonequilibrium crystallization.

With increasing laser power, the HAZ width grows and the microstructure becomes coarser. Thus, the size of crystals in the HAZ comes close to 15 μm for sample No. 1, and 100 μm for sample No. 5, while the base metal (under the HAZ) is characterized by a ferrite-pearlite microstructure with grain sizes of about 10–15 μm.

Since the coating technology uses thermal effects, it can be assumed that a diffusion layer can appear at the interface between the steel substrate and the bronze coating with the formation of solid solutions that occur in the Cu–Al–Fe system [22,23,24].

The presence of a diffusion zone (see Figure 7) should facilitate the adhesion of the coating material to the substrate material, thereby providing good adhesion properties. The width of the diffusion zone in tested samples depends on the laser power and is about 5 μm for sample No. 1 and 30 μm for sample No. 5 (see Figure 7). Such values of the diffusion zone width are higher than that of a coating applied by electric arc method: according to Zhang et al. [7], copper diffusion from the bronze coating to steel was about 3 μm while aluminum diffusion did not exceed 10 μm.

For all samples, the average microhardness of the coating is higher than that of the coating material itself (the HV of the aluminum bronze used is about 100) and higher than that of the substrate material (the HV of the structural steel used is 202), which is typical for composite materials.

According to Alam et al. [8], the microhardness of aluminum bronze coating applied to steel by plasma spraying varied from 100 to 160 HV depending on the spraying conditions, which was on average one and a half times less than for our test samples. Furthermore, Freiße et al. [11] shown that the microhardness of the coating of the same grade of aluminum bronze deposited by laser on steel, averaged 200–220 HV.

The dry friction coefficient for the studied samples was 0.389–0.574, which is several times higher than for the samples coated with plasma spraying (according to Alam et al. [8], the friction coefficient at a load of 100 N ranged from 0.02 to 0.08). However, our data are comparable with those obtained by Freiße et al. [11], according to which the dry friction coefficient varied from 0.58 to 0.79 depending on the parameters of the applied surfacing for the laser–surfaced bronze Cu–10Al–1Fe coating on steel.

## 5. Conclusions

The results of the studies allow the following conclusions:(1)The microstructure of the aluminum bronze coating applied to steel using additive techniques is typical of a composite material.(2)An increase in laser power makes the coating profile dense, smooth, and more even in thickness, while the microstructure becomes more uniform, but at the same time, coarser.(3)The thermal effect during coating promotes the diffusion of copper and aluminum from bronze to steel with the formation of a diffusion zone 5–30 μm wide, which should facilitate adhesion of the coating material to the substrate material.(4)When surfacing, overheating occurs in the heat-affected zone in the substrate, which leads to the formation of a layer with the Widmanstätten microstructure. The depth of such an overheated layer depends on the laser power.(5)The microhardness of the applied coating is 1.5–2.5 times higher than the microhardness of aluminum bronze or the used steel grade in its pure form.(6)The absolute values of hardness over the thickness of the coating are consistent with its microstructure, that is, fluctuations are observed.(7)According to the results of tribological tests, this coating cannot be considered as antifriction, but rather as a promising material for coating, for example, brake pads.

## Figures and Tables

**Figure 1 materials-13-00461-f001:**
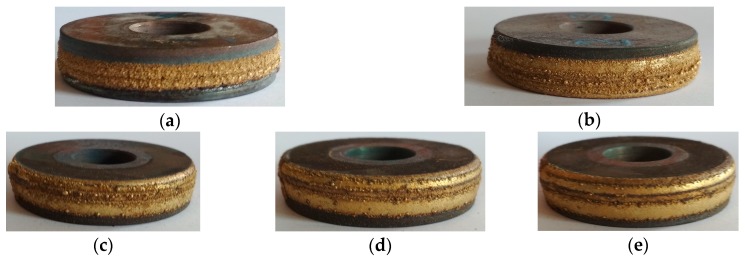
The appearance of the bronze coating obtained for samples: (**a**) No. 1; (**b**) No. 2; (**c**) No. 3; (**d**) No. 4; (**e**) No. 5. The outer diameter of all the rollers is 45 mm.

**Figure 2 materials-13-00461-f002:**
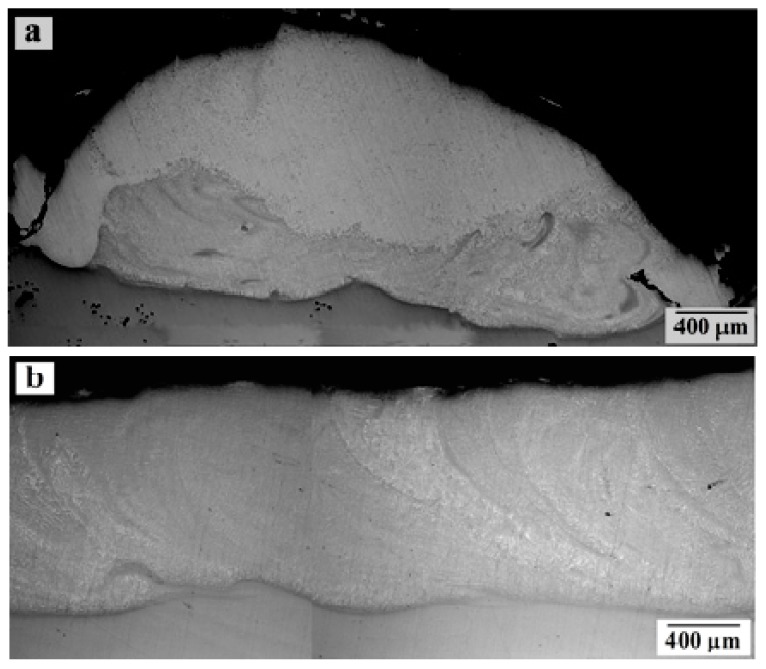
Coating profiles according to optical microscopy for samples: (**a**) No. 2 and (**b**) No. 5.

**Figure 3 materials-13-00461-f003:**
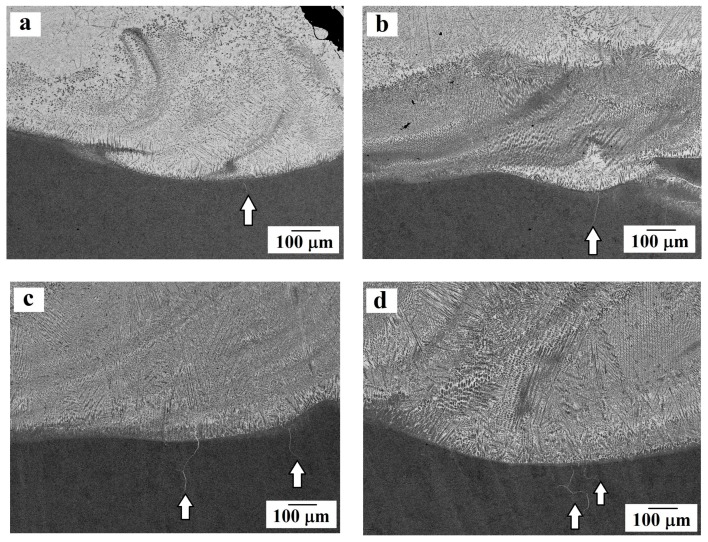
The microstructure of the substrate-coating contact zone (according to SEM data). General views for samples: (**a**) No. 2; (**b**) No. 3; (**c**) No. 4; (**d**) No. 5. Arrows indicate microcracks filled with bronze.

**Figure 4 materials-13-00461-f004:**
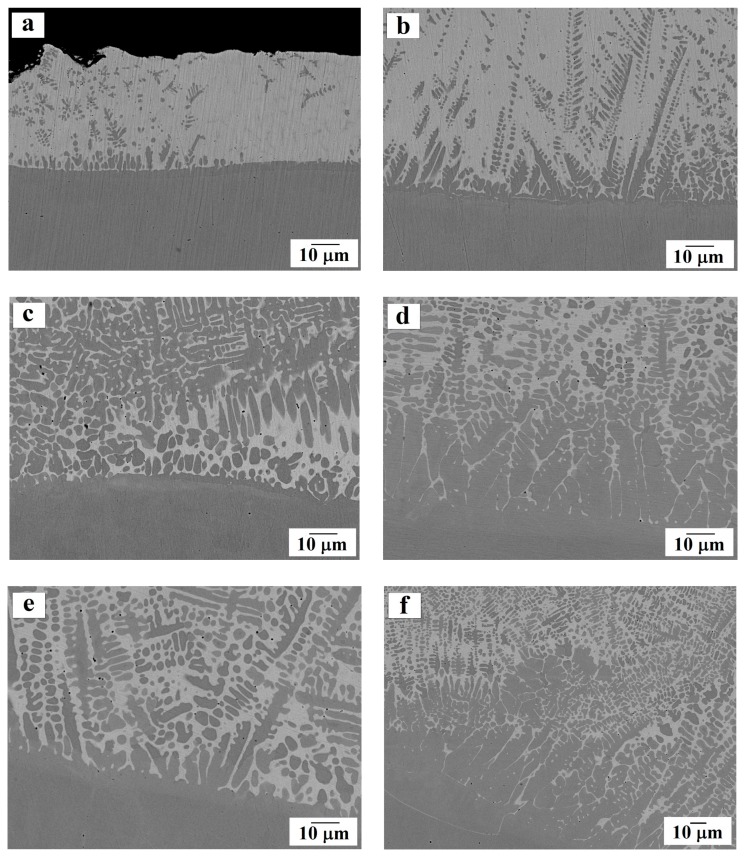
The microstructure of the substrate-coating contact zone (according to SEM data). Magnified fragments for samples: (**a**) No. 1; (**b**) No. 2; (**c**) No. 3; (**d**) No. 4; (**e**) and (**f**) No. 5.

**Figure 5 materials-13-00461-f005:**
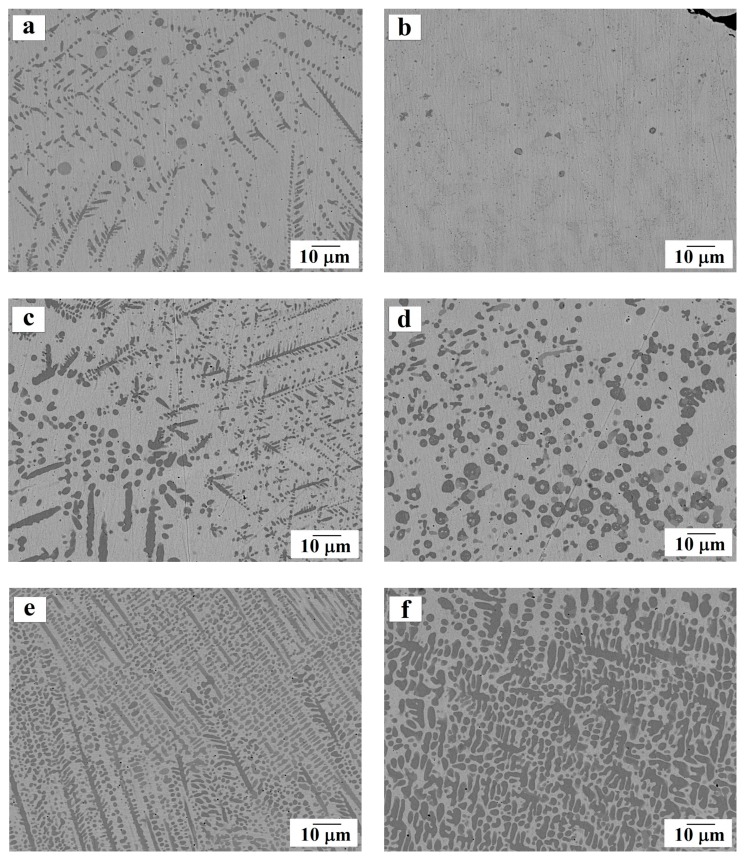
The microstructure of the central part of the obtained coating (according to SEM data) for samples: (**a**) and (**b**) No. 2; (**c**) and (**d**) No. 3; (**e**) No. 4; (**f**) No. 5.

**Figure 6 materials-13-00461-f006:**
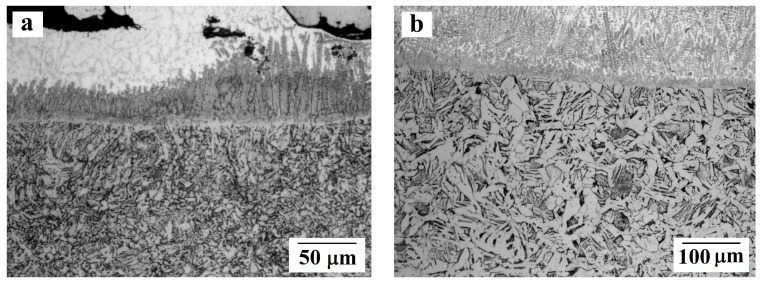
The HAZ microstructure (according to optical microscopy) for samples: (**a**) No. 1; (**b**) No. 5.

**Figure 7 materials-13-00461-f007:**
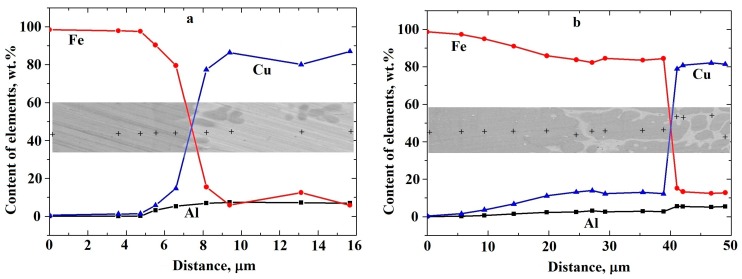
The results of the X-ray microanalysis for the contact zone of sample: (**a**) No. 1; (**b**) No. 5.

**Table 1 materials-13-00461-t001:** Manufacturing parameters of coating deposition.

No.	Number of Cycles	Laser Power (W)	Laser Beam Travel Speed (mm/s)	Spot Diameter (mm)	Powder Feed Rate (g/min)
1	2	600	12	2	15
2	2	1000			
3	2	1400			
4	1	1800			
5	1	2200			

**Table 2 materials-13-00461-t002:** Average compositions of the obtained composite coating (according to XRMA), weight %.

No.	Al	Si	Mn	Fe	Cu
1	7.49	0.07	0.11	12.87	79.46
2	7.71	0.09	0.12	18.25	73.83
3	7.96	0.09	0.06	24.25	67.64
4	5.91	0.11	0.20	35.22	58.56
5	4.98	0.14	0.31	44.95	49.62

**Table 3 materials-13-00461-t003:** The dry friction coefficient and wear rate (the average at the three different test loads) for the test sample coatings.

No.	Dry Friction Coefficient *			Wear Rate
	200 N	300 N	400 N	
2	0.389	0.537	0.545	(6.96 ± 0.74) × 10^–5^
3	0.422	0.563	0.563	(5.79 ± 0.65) × 10^–5^
4	0.539	0.548	0.568	(4.62 ± 0.48) × 10^–5^
5	0.541	0.548	0.574	(4.25 ± 0.41) × 10^–5^

* Measurement error did not exceed 5%.

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
