# Peer review of "A Study of Characteristics of Aluminum Bronze Coatings Applied to Steel Using Additive Technologies"

_materials, 2020, doi:10.3390/ma13020461_

Round 1

Reviewer 1 Report

1. Suggestion for the presented microstructure photos Fig. 3-5: Separate the photos with a horizontal gap of a few millimeters so that the microstructure does not mix - it would be better for the reader.
2. I lacked some information, namely: is the remelting laser a carbon dioxide laser? What was the laser mode of heating -
continuous or impulse?
3. What was the trajectory of the laser paths: spiral or circle? Did the melted paths touch each other or did they overlap to some extent ?.
4. Impressive remelting results as the laser power increases. Definitely sufficient diffusion zone thickness for sample No. 5, ensuring good coherence of the layer with the material core.
5. The dimensions of dendrites, enlarged needles similar to the microstructure of Widmanssteten are a little disturbing. In this case, I would suggest the use of laser pulse operation in further studies, which allows for a multiple increase of energy in 1/3 of the pulse duration, and in 2/3 of the energy pulse duration much lower than the enerard. on impulse. This changes the nature of heating and cooling and can affect the microstructure fineness.
6. The results are very interesting, in the future it would be possible to improve the methodology of the experiment using rototable plans and obtain a description of the results in the form of a continuous function, e.g. second-degree with interactions.

Author Response

Dear Reviewer, first of all, many thanks for the careful attitude to our manuscript.

Thank you for your valuable suggestions and comments.

Let us answer your сomments and suggestions.

Suggestion for the presented microstructure photos Fig. 3-5: Separate the photos with a horizontal gap of a few millimeters so that the microstructure does not mix - it would be better for the reader.

Answer. Thank you very much for the recommendation, we have made changes to the text of the manuscript.

I lacked some information, namely: is the remelting laser a carbon dioxide laser? What was the laser mode of heating -continuous or impulse?

Answer. 4 kW laser head with an ytterbium fiber-optics laser (LS-4). The heating mode is continuous. Corresponding corrections have been made to the text of the manuscript.

What was the trajectory of the laser paths: spiral or circle? Did the melted paths touch each other or did they overlap to some extent ?

Answer. The trajectory of laser path was linear. Surfaced strips of coating partially overlaped each other. The overlap width was 1.8 mm. Corresponding corrections have been made to the text of the manuscript.

Impressive remelting results as the laser power increases. Definitely suff icient diffusion zone thickness for sample No. 5, ensuring good coherence of the layer with the material core.

Answer. Thanks. You are right, the results are very interesting.

The dimensions of dendrites, enlarged needles similar to the microstructure of Widmanssteten are a little disturbing. In this case, I would suggest the use of laser pulse operation in further studies, which allows for a multiple increase of energy in 1/3 of the pulse duration, and in 2/3 of the energy pulse duration much lower than the enerard. on impulse. This changes the nature of heating and cooling and can affect the microstructure fineness.

Answer. You are certainly right. The rough Widmanstätt-like structure is not very technologically advanced. We will take into account your recommendations in future work.

The results are very interesting, in the future it would be possible to improve the methodology of the experiment using rototable plans and obtain a description of the results in the form of a continuous function, e.g. second-degree with interactions.

Answer. Thanks. We will take into account your recommendations in future work.

Reviewer 2 Report

INTRODUCTION

Lines 33-34: the terms used in these lines, e.g. “details” and “friction units”, are not clear and possibly incorrect. Do the authors intend to refer, generically, to mechanical components and constituents of a tribological coupling, or do they want to indicate some specific component? Line 51: what do the authors mean by “dispersed” microstructure? Do they mean they consist of a uniform dispersion of fine secondary phases? Lines 51-52: the concept of “loss on abrasion and friction coefficient” is not very clear. Do the authors mean that the specific wear rate and friction coefficient of the plasma sprayed coating had a sudden transition during testing?

SECTION 2

Lines 61-63: please specify the particle size distribution of the feedstock powder and, if possible, please provide an SEM micrograph of the same. Line 63: please specify the size of the substrate. When describing the laser cladding system (line 66), please specify the type of laser source and its wavelength. It is not clear whether the samples were covered by complete coatings or single laser scans. In the former case, please specify the spacing between adjacent passes and, therefore, the % overlap (considering the laser spot diameter).

In addition, please describe briefly the powder feeding system (i.e. whether it is coaxial, lateral, …) and please specify whether cladding was carried out under shielding gas. If so, please provide the shielding gas type and flow rate.

Line 76-79: the contact geometry in the tribological tests is not clearly inferred from the description given in these lines. If possible, please provide a schematic of the device and specify the geometry and size of the coated samples.

Please also specify the width of the roller, the surface roughness of the roller and coated samples, and some key process parameters including normal load and test duration.

How did the authors measure the wear volume loss? Please provide some experimental details in the text.

SECTION 3.1

Line 90: the thickness value of coating 1 is lower by one order of magnitude than that of all other coatings. Please check whether the value was printed correctly. Line 98: please specify the type of etchant. It is not clear at which position across the coating thickness each of the images of Fig. 5 was taken. It might be advisable to provide, for each sample, a sequence of micrographs showing the complete transition from the outer surface down to the heat-affected zone. By the way, please note that the term “heat-affected zone” (HAZ) is more frequently used in the literature than the term “zone of thermal effect” (ZTE).

SECTION 3.3

I would recommend providing a graph with the through-thickness hardness trend of all samples. Please also specify the error ranges associated with the hardness measured at each location across the sample.

The text in Section 2 mentions only 10 indentations per sample, but, if the authors performed measurements at different depths, the number of indentations should probably be larger than that. Multiple indentations (at least 5 – the more, the better) should, indeed, have been performed at each depth level, in order to get a minimum of statistical significance.

SECTION 3.4

In Table 3, please provide error ranges associated to each average value (e.g. the maximum half-difference between the values measured in the three repeats of each test, as mentioned on line 79). In Table 3, only one wear rate is provided. To what applied load does it correspond? Would it be possible to provide wear rates at every load?

SECTION 4

Lines 169-170: the mention of a “metastable state of the liquid phase” is not very clear. Do the authors mean that the liquid becomes supercooled?

In general, if the authors assumed some non-equilibrium process took place, its mechanism should be described in some more detail in the text.

Lines 173-175: the mention to bronze-filled cracks is somewhat puzzling, as those cracks were not mentioned in the Section 3.1, and are not easily identifiable in the optical and SEM micrographs. Could the authors please point them out in the figures, mention them in the text of Section 3.1, and provide evidence to their bronze filling? Lines 190-195: it would be advisable to clarify where the diffusion layer described in these lines and shown in Fig. 7 is located with respect to the heat-affected zone described in the previous lines and shown in Fig. 6. Does the heat-affected zone begin where interdiffusion of steel with Cu ends, or do the Widmanstätten structures described in Fig. 6 also encompass Cu-enriched parts? The mentioned steel composition, which Cr, Ni and Mo as alloying elements, would seem to be well suited for a quenching+tempering process. This might have raised the hardness well above the ≈200 HV level mentioned in the text. Is there a specific technological reason why the present substrates were in an annealed (ferritic/pearlitic) state, instead? The tribological discussion is extremely limited. It would be advisable to expand this part significantly, showing SEM micrographs of worn samples and counterparts in order to determine wear mechanisms. Another significant shortcoming of the present tribological tests is the lack of terms of comparison for the measured wear rates. Since wear loss is a comparative (rather than absolute) measure, one or more terms of comparison (e.g. uncoated steel, bulk Al-bronze, …) should also be tested.

Throughout the paper, please note that there seems to be a confusion between the terms “structure” (which describes the atomic arrangement into amorphous or crystalline features) and “microstructure” (which indicates the distribution of phases and crystalline grains in a material). What the authors describe throughout the text is, in fact, the microstructure of the coatings, not their structure.

Please also refer to the attached pdf file for additional suggested language and typing corrections.

Author Response

Dear Reviewer, first of all, many thanks for the careful attitude to our manuscript.

Thank you for your valuable suggestions and comments.

Let us answer your сomments and suggestions.

Lines 33-34: the terms used in these lines, e.g. “details” and “friction units”, are not clear and possibly incorrect. Do the authors intend to refer, generically, to mechanical components and constituents of a tribological coupling, or do they want to indicate some specific component?

Answer. In this context, there is no specific reference to specific parts in machines units. Since we use a general phrase to describe all possible friction pairs.

Line 51: what do the authors mean by “dispersed” microstructure? Do they mean they consist of a uniform dispersion of fine secondary phases?

Answer. By the term "dispersed microstructure" we meant the small size of the structural components. Corresponding corrections have been made to the text of the manuscript.

Lines 51-52: the concept of “loss on abrasion and friction coefficient” is not very clear. Do the authors mean that the specific wear rate and friction coefficient of the plasma sprayed coating had a sudden transition during testing?

Answer. You are absolutely right, the authors of the work [12], to which we refer, were meant it and wrote the corresponding results.

Lines 61-63: please specify the particle size distribution of the feedstock powder and, if possible, please provide an SEM micrograph of the same.

Answer. As the coating material, we chose aluminum bronze powder (particles with sizes from 40 to 100 micrometers). Corresponding corrections have been made to the text of the manuscript.

Line 63: please specify the size of the substrate. When describing the laser cladding system (line 66), please specify the type of laser source and its wavelength. It is not clear whether the samples were covered by complete coatings or single laser scans. In the former case, please specify the spacing between adjacent passes and, therefore, the % overlap (considering the laser spot diameter).

Answer. The outer diameter of all the rollers (substrates) is 45 mm. 4 kW laser head with an ytterbium fiber-optics laser with wavelength 1065–1075 nm (LS-4), the heating mode is continuous. The trajectory of laser path was linear. Surfaced strips of coating partially overlaped each other. The overlap width was 1.8 mm. Corresponding corrections have been made to the text of the manuscript.

In addition, please describe briefly the powder feeding system (i.e. whether it is coaxial, lateral, …) and please specify whether cladding was carried out under shielding gas. If so, please provide the shielding gas type and flow rate.

Answer. We used TWIN-10-CR-2 powder feeder with a four-axis powder feed module. The coating was carried out under shielding gas (argon), the flow rate of which was 20-22 L/min. Corresponding corrections have been made to the text of the manuscript.

Line 76-79: the contact geometry in the tribological tests is not clearly inferred from the description given in these lines. If possible, please provide a schematic of the device and specify the geometry and size of the coated samples.

and

Please also specify the width of the roller, the surface roughness of the roller and coated samples, and some key process parameters including normal load and test duration.

and

How did the authors measure the wear volume loss? Please provide some experimental details in the text.

Answer. Tribological tests were carried out on the II-5018 friction machine (Russia) according to the scheme “movable roller – stationary roller (test material)”. The diameter of both rollers was 45 mm; the thickness was 10 mm. Tribological tests were carried out at room temperature. Before testing, the friction surface was ground on a abrasive material with a grain size of 9 μm. Hardness of counter material was 490 HV. Wear loss was determined by changing the thickness of the coating. The test time was determined by the length of the path traveled, which for the test samples was 100 m. Wear rate was defined as the ratio of wear loss to the distance traveled. Corresponding corrections have been made to the text of the manuscript.

Line 90: the thickness value of coating 1 is lower by one order of magnitude than that of all other coatings. Please check whether the value was printed correctly.

Answer. Indeed, with the low power of the laser used, the coating is quite thin (rechecked several times). Also, the full thickness of the coating can be seen in Fig. 4a.

Line 98: please specify the type of etchant.

Answer. The substrate material was etched with a 4% solution of nitric acid in ethanol. Corresponding corrections have been made to the text of the manuscript.

It is not clear at which position across the coating thickness each of the images of Fig. 5 was taken. It might be advisable to provide, for each sample, a sequence of micrographs showing the complete transition from the outer surface down to the heat-affected zone.

Answer. Figure 5 shows the structure of the central part of the obtained coating. Corresponding corrections have been made to the text of the manuscript.

By the way, please note that the term “heat-affected zone” (HAZ) is more frequently used in the literature than the term “zone of thermal effect” (ZTE).

Answer. Thanks! We did not know the exact terminology in English; corresponding corrections have been made to the text of the manuscript.

I would recommend providing a graph with the through-thickness hardness trend of all samples. Please also specify the error ranges associated with the hardness measured at each location across the sample.

Answer. After measuring the microhardness, we tried to present the results in graphical form, however, as indicated in the text of the manuscript, the results have a large scatter due to the observed microstructure. Therefore, the graphics turned out to be sawtooth and uninformative. Therefore, it was decided to indicate the range of hardness values for each coating.

The text in Section 2 mentions only 10 indentations per sample, but, if the authors performed measurements at different depths, the number of indentations should probably be larger than that. Multiple indentations (at least 5 – the more, the better) should, indeed, have been performed at each depth level, in order to get a minimum of statistical significance.

Answer. The number of microhardness measurements varied from 10 to 100 both across and along the obtaining coating (the number of measurements depended on the thickness of the coating). Corresponding corrections have been made to the text of the manuscript.

In Table 3, please provide error ranges associated to each average value (e.g. the maximum half-difference between the values measured in the three repeats of each test, as mentioned on line 79). In Table 3, only one wear rate is provided. To what applied load does it correspond? Would it be possible to provide wear rates at every load?

Answer. Changes in load did not significantly affect on the wear rates. All three values were comparable with each other, so we decided to indicate in the table for simplicity of perception only the average value.

Lines 169-170: the mention of a “metastable state of the liquid phase” is not very clear. Do the authors mean that the liquid becomes supercooled?

Answer. Yes. Here again, there are subtleties of the translation of terms.

In general, if the authors assumed some non-equilibrium process took place, its mechanism should be described in some more detail in the text.

Answer. Since the use of a laser involves accelerated heating and accelerated cooling of the metal (due to heat removal to the environment and the substrate), it is obvious that the processes occur in non-equilibrium conditions.

Lines 173-175: the mention to bronze-filled cracks is somewhat puzzling, as those cracks were not mentioned in the Section 3.1, and are not easily identifiable in the optical and SEM micrographs. Could the authors please point them out in the figures, mention them in the text of Section 3.1, and provide evidence to their bronze filling?

Answer. Corresponding corrections have been made to the text of the manuscript.

Lines 190-195: it would be advisable to clarify where the diffusion layer described in these lines and shown in Fig. 7 is located with respect to the heat-affected zone described in the previous lines and shown in Fig. 6. Does the heat-affected zone begin where interdiffusion of steel with Cu ends, or do the Widmanstätten structures described in Fig. 6 also encompass Cu-enriched parts?

Answer. As said in the discussion (see section 4), the thickness of the diffusion zone does not exceed 30 micrometer.

The mentioned steel composition, which Cr, Ni and Mo as alloying elements, would seem to be well suited for a quenching+tempering process. This might have raised the hardness well above the ≈200 HV level mentioned in the text. Is there a specific technological reason why the present substrates were in an annealed (ferritic/pearlitic) state, instead?

Answer. In the course of the work, the hardness of the coating was measured directly, but the hardness of the steel substrate was not. The indicated parameter is given for steel in the initial (uncoated) annealed state. Corresponding corrections have been made to the text of the manuscript.

The tribological discussion is extremely limited. It would be advisable to expand this part significantly, showing SEM micrographs of worn samples and counterparts in order to determine wear mechanisms. Another significant shortcoming of the present tribological tests is the lack of terms of comparison for the measured wear rates. Since wear loss is a comparative (rather than absolute) measure, one or more terms of comparison (e.g. uncoated steel, bulk Al-bronze, …) should also be tested.

Answer. Thanks for the recommendation. We will definitely carry out the studies you recommend in future work.

Throughout the paper, please note that there seems to be a confusion between the terms “structure” (which describes the atomic arrangement into amorphous or crystalline features) and “microstructure” (which indicates the distribution of phases and crystalline grains in a material). What the authors describe throughout the text is, in fact, the microstructure of the coatings, not their structure.

Answer. Corresponding corrections have been made to the text of the manuscript.

Please also refer to the attached pdf file for additional suggested language and typing corrections.

Answer. We hope our answers satisfied you. In addition, we want to express you great gratitude for editing our manuscript and the provided corrected PDF file. Corresponding corrections have been made to the text of the manuscript.

Reviewer 3 Report

Here is the list of my comments:

-The wear test it must be explained in more detail. Also, the wear rate should be explained how was calculated and the results should be included in the discussion

- It is used the term Zone of thermal Effect, this thermology it is not used, it should be change by Heat affected Zone, known as HAZ

Author Response

Dear Reviewer, first of all, many thanks for the careful attitude to our manuscript.

Thank you for your valuable suggestions and comments.

Let us answer your сomments and suggestions.

The wear test it must be explained in more detail. Also, the wear rate should be explained how was calculated and the results should be included in the discussion

Answer. Tribological tests were carried out at room temperature. Before testing, the friction surface was ground on a abrasive material with a grain size of 9 μm. Hardness of counter material was 490 HV. Wear loss was determined by changing the thickness of the coating. The test time was determined by the length of the path traveled, which for the test samples was 100 m. Wear rate was defined as the ratio of wear loss to the distance traveled. Corresponding corrections have been made to the text of the manuscript.

It is used the term Zone of thermal Effect, this thermology it is not used, it should be change by Heat affected Zone, known as HAZ

Answer. Thanks! We did not know the exact terminology in English; corresponding corrections were made to the text of the manuscript.

Reviewer 4 Report

This paper studies the microstructure, hardness, and sliding wear properties of aluminium bronze coatings laser-clad to the low alloy steel. Aluminium bronze coatings are frequently used in various sliding wear applications due to low friction and moderate wear characteristics. Laser cladding is a low heat input overlay welding method, which use in industrial applications is increasing. Therefore, I recommend to publish this manuscript after revisions. Detailed comments below:

Page 1, Introduction, Line 41: 'laser beam diameter'

Page 2, Materials and Methods: particle size distribution of powder must be given, dimensions of the substrate must be given, preparation of substrate before cladding must be given, heat-treatment condition of the substrate must be given (hardened, quenched&tempered etc.?)

Page 2, Materials and Methods: laser metal-cladding unit must be described more thoroughly, laser type, max powder, fiber diameter, collimator, optics, nozzle configuration (off-axis, co-axial), powder feeder, robotics etc.

Page 2, Materials and Methods: single beads or multiple beads? Overlapping rate? single-layer or multi-layer? Explain the number of cycles in Table 1.

Page 2, Materials and Methods: What was the temperature in the wear tests? Hardness of counter material 5140H? Contact stress at the beginning of the test? Test time? How the surfaces were prepared? Surface condition/roughness in the beginning? How the wear loss was measured?

Page 3, Results, Figure 1: scale bar or dimensions of discs must be given.

Page 4, Results, Line 100: ZTE is commonly known as a heat-affected zone (HAZ). Use that throughout the paper.

Page 6, Results, Line 135: Tribological investigations.

Page 6, Results, Table 3: Explain how the wear rate was calculated.

Page 7, Discussion, Lines 176-182: Due to high carbon equivalent of the substrate and fast cooling, HAZ just simply in other words hardened, right?

Author Response

Dear Reviewer, first of all, many thanks for the careful attitude to our manuscript.

Thank you for your valuable suggestions and comments.

Let us answer your сomments and suggestions.

Page 1, Introduction, Line 41: 'laser beam diameter'.

Answer. Corresponding corrections have been made to the text of the manuscript.

Page 2, Materials and Methods: particle size distribution of powder must be given, dimensions of the substrate must be given, preparation of substrate before cladding must be given, heat-treatment condition of the substrate must be given (hardened, quenched&tempered etc.?)

Answer. As the coating material, we chose aluminum bronze powder (particles with sizes from 40 to 100 micrometers). The outer diameter of all the rollers (substrates) is 45 mm. The substrate material was used in the annealed state. Corresponding corrections have been made to the text of the manuscript.

Page 2, Materials and Methods: laser metal-cladding unit must be described more thoroughly, laser type, max powder, fiber diameter, collimator, optics, nozzle configuration (off-axis, co-axial), powder feeder, robotics etc.

Answer. The main unit of the complex can be represented as follows: 1) 4 kW laser head with an ytterbium fiber-optics laser with wavelength 1065–1075 nm (LS-4), the heating mode is continuous; 2) KUKA R-120 six-axis robot-equipped manipulator combined with KUKA DKP-400 double-axis positioning element; 3) TWIN-10-CR-2 powder feeder with a four-axis powder feed module; 4) process chamber – a metal cylinder with the diameter of 600 mm and the length of 1100 mm. Corresponding corrections have been made to the text of the manuscript.

Page 2, Materials and Methods: single beads or multiple beads? Overlapping rate? single-layer or multi-layer? Explain the number of cycles in Table 1.

Answer. The trajectory of laser path was linear. Surfaced strips of coating partially overlaped each other. The overlap width was 1.8 mm. The number of cycles in the Table 1 corresponds to the number of laser passes providing the required coating thickness.. For samples No. 1–3, due to the unevenness of the obtained coating, two cycles of laser surfacing were performed. Corresponding corrections have been made to the text of the manuscript.

Page 2, Materials and Methods: What was the temperature in the wear tests? Hardness of counter material 5140H? Contact stress at the beginning of the test? Test time? How the surfaces were prepared? Surface condition/roughness in the beginning? How the wear loss was measured?

Answer. Tribological tests were carried out at room temperature. Before testing, the friction surface was ground on a abrasive material with a grain size of 9 μm. The test time was determined by the length of the path traveled, which for the test samples was 100 m. Hardness of counter material was 490 HV. Wear loss was determined by changing the thickness of the coating. Corresponding corrections have been made to the text of the manuscript.

Page 3, Results, Figure 1: scale bar or dimensions of discs must be given.

Answer. Corresponding corrections have been made to the text of the manuscript.

Page 4, Results, Line 100: ZTE is commonly known as a heat-affected zone (HAZ). Use that throughout the paper.

Answer. Thanks! We did not know the exact terminology in English; corresponding corrections were made to the text of the manuscript.

Page 6, Results, Line 135: Tribological investigations.

Answer. Corresponding corrections have been made to the text of the manuscript.

Page 6, Results, Table 3: Explain how the wear rate was calculated.

Answer. Wear rate was defined as the ratio of wear loss to the distance traveled. Corresponding corrections have been made to the text of the manuscript.

Page 7, Discussion, Lines 176-182: Due to high carbon equivalent of the substrate and fast cooling, HAZ just simply in other words hardened, right?

Answer. In the course of the work, the hardness of the coating was measured directly, but the hardness of the steel substrate was not.

Round 2

Reviewer 2 Report

Replies to previous comments are mostly clear and appropriate.

I would still ask the authors to perform few additional changes:

1) I would still recommend that error ranges (expressed e.g. as half-difference between extreme values, or as standard error, depending on the number of experimental values that were averaged together) be provided for friction coefficients and wear rates in Table 3. The average value of an experimental measurement, indeed, has limited significance, if the associated uncertainty is not known.

2) The reply clarifies that the wear rate values in Table 3 are the average of results at the three different test loads. This, however, is nowhere specified in the text. I would therefore recommend that this be stated explicitly in Section 3.4 and/or in the table caption.

3) A few additional typing corrections can be recommended as follows:

l. 51: have smaller size of the structural components -> consists of finer microstructural components l. 86: the obtaining coating -> the obtained coating l. 93: a abrasive -> an abrasive l. 94: was determined by changing the thickness of the coating -> was determined by measuring the thickness loss of the coating l. 189-190: Such precipitates are typical for processes of metastable state of the liquid phase -> Such precipitates are typical of processes involving a supercooled liquid phase

Author Response

Dear Reviewer, first of all, many thanks for the careful attitude to our manuscript.

Thank you for your valuable suggestions and comments.

Let us answer your сomments and suggestions.

I would still recommend that error ranges (expressed e.g. as half-difference between extreme values, or as standard error, depending on the number of experimental values that were averaged together) be provided for friction coefficients and wear rates in Table 3. The average value of an experimental measurement, indeed, has limited significance, if the associated uncertainty is not known.

Answer. Thank you for your comment. We have added the necessary refinement to the Table 3.

The reply clarifies that the wear rate values in Table 3 are the average of results at the three different test loads. This, however, is nowhere specified in the text. I would therefore recommend that this be stated explicitly in Section 3.4 and/or in the table caption.

Answer. Thank you for your comment. We have added the necessary refinement to the Table 3 header.

3) A few additional typing corrections can be recommended as follows:

51: have smaller size of the structural components -> consists of finer microstructural components l. 86: the obtaining coating -> the obtained coating l. 93: a abrasive -> an abrasive l. 94: was determined by changing the thickness of the coating -> was determined by measuring the thickness loss of the coating l. 189-190: Such precipitates are typical for processes of metastable state of the liquid phase -> Such precipitates are typical of processes involving a supercooled liquid phase

Answer. Thank you very much for editing the English text. We have made changes to the text of the manuscript.

Team of Authors

Reviewer 3 Report

I agrre the cahnges realised.

Author Response

Dear Reviewer.

 Once again, we want to thank you for your careful attitude to our manuscript and valuable recommendations for improving it.

Team of Authors.

Reviewer 4 Report

Some text editing still needed:

Page 2,line 74: overlaped -> overlapped

Page 8, line 187: occup -> occupies

Page 9, line 231: '...conforms is...'. please check this.

Author Response

Dear Reviewer.

We would like to once again thank you for your attentive attitude to our maniscript and for the valuable comments made and recommendations for its improvement.

According to recent comments, we have made changes to the text.

Team of Authors